# Automatic Sleep Disorders Classification Using Ensemble of Bagged Tree Based on Sleep Quality Features

**Edita Rosana Widasari [1]** , **Koichi Tanno [2,]**\* and **Hiroki Tamura [3]**

[1]  Interdisciplinary Graduate School of Agriculture and Engineering, Department of Material and Informatics, University of Miyazaki, Miyazaki 889-2192, Japan; editarosanaw@gmail.com
[2]  Department of Electrical and System Engineering, University of Miyazaki, Miyazaki 889-2192, Japan
[3]  Department of Environmental Robotics, University of Miyazaki, Miyazaki 889-2192, Japan; htamura@cc.miyazaki-u.ac.jp
\*  Correspondence: tanno@cc.miyazaki-u.ac.jp

**Abstract:** Sleep disorder is a medical disease of the sleep patterns, which commonly suffered by the elderly. Sleep disorders diagnosis and treatment are considered to be challenging due to a time-consuming and inconvenient process for the patient. Moreover, the use of Polysomnography (PSG) in sleep disorder diagnosis is a high-cost process. Therefore, we propose an efficient classification method of sleep disorder by merely using electrocardiography (ECG) signals to simplify the sleep disorders diagnosis process. Different from many current related studies that applied a five-minute epoch to observe the main frequency band of the ECG signal, we perform a pre-processing technique that suitable for the 30-seconds epoch of the ECG signal. By this simplification, the proposed method has a low computational cost so that suitable to be implemented in an embedded hardware device. Structurally, the proposed method consists of five stages: (1) pre-processing, (2) spectral features extraction, (3) sleep stage detection using the Decision-Tree-Based Support Vector Machine (DTB-SVM), (4) assess the sleep quality features, and (5) sleep disorders classification using ensemble of bagged tree classifiers. We evaluate the effectiveness of the proposed method in the task of classifying the sleep disorders into four classes (insomnia, Sleep-Disordered Breathing (SDB), REM Behavior Disorder (RBD), and healthy subjects) from the 51 patients of the Cyclic Alternating Pattern (CAP) sleep data. Based on experimental results, the proposed method presents 84.01% of sensitivity, 94.17% of specificity, 86.27% of overall accuracy, and 0.70 of Cohen's kappa. This result indicates that the proposed method able to reliably classify the sleep disorders merely using the 30-seconds epoch ECG in order to address the issue of a multichannel signal such as the PSG.

**Keywords:** sleep disorders in the elderly; ECG signal; sleep stage; DTB-SVM; sleep quality; ensemble of bagged tree

## 1. Introduction

The sleep disorder is a medical disease of the sleep patterns of a person. Naturally, the elderly are easier to experience sleep disorders compared to younger people. There are two types of sleep disorders that commonly suffer the elderly, i.e., insomnia and primary sleep disorders [1–3]. Insomnia is typically defined as a difficulty for falling asleep. It affected almost 40%–50% of the elderly [2]. Furthermore, primary sleep disorders are the other sleep disorder that not attributable to a psychiatric condition, such as sleep-disordered breathing (SDB), REM Behavior Disorder (RBD), and Restless Legs Syndrome (RLS).



The prevalence of SDB in the elderly is about 20%–40% [4]. While based on the RBD questionnaire, the prevalence of RBD in the elderly is 4.6%–7.7%. It will be much higher for certain neurodegenerative diseases [5]. A study estimated that there are 9%–20% of the elderly with RLS. However, this estimation may include a substantial portion of the Periodic Limb Movement of Sleep (PLMS) patients. It is because RLS is often related to PLMS. Approximately, 70% of the patients with RLS also have PLMS, but only 20% of the patients with PLMS are reported RLS. Thus, we argue that the consideration of RLS is presenting a different diagnosis to any patient with PLMS [2]. Therefore, we consider that insomnia, SDB, and RBD could represent the most common sleep disorder in the elderly.

Polysomnography (PSG) is the standard method used to diagnose sleep disorders. The PSG method involves a lot of wired sensors to record the activities of the multiple physiological signals (such as brain waves, skeletal muscles, heart rate, eye movement, etc.). This method is conducted for all overnight long in a specialized laboratory or hospital. Moreover, misdiagnosis and mistreatment may occur though the patients had been monitored as long as full sleep observation. This observation method makes the patients feel inconvenience so that many patients refused the observation since they consider this method as need an extra effort, time-consuming, and high-cost [6]. It makes the elderly with sleep disorders are undiagnosed and untreated in clinical practice.

In the last decade, a clinical study [7] mentioned that it is possible to detect a sleep stage and sleep disorder in the elderly using an Electrocardiogram (ECG) signal instead of complicated signal recordings. It is because each sleep stage has different cardiac dynamics, which are represented in the average of the heartbeat interval [8]. To estimate the sleep stage, we can use the differences between the Autonomic Nervous System (ANS) activities from Heart Rate Variability (HRV) signal [9]. HRV performs a fluctuation analysis in the heartbeat interval so that the variation of HRV is according to the sleep stage, and reflect the activity of ANS.

To this end, we propose an efficient non-intrusive method to classify sleep disorders automatically for the elderly using the ECG signal alone. This method offers a more comfortable method than the conventional method but able to provide the result as effective as the PSG method. In addition, the advantage of ECG is more convenient to used and recorded, although without specialized trainers. A key process of our proposed method is the selection techniques of the pre-processing stage with regards to decompose the 30-seconds epoch of the ECG signal. Furthermore, we used a supervised machine learning in the task of classification. We perform two approaches: (1) sleep stage detection using Decision-Tree-Based Support Vector Machine (DTB-SVM) based on spectral features of ECG signal, and (2) sleep disorders classification using ensemble of bagged tree based on sleep quality features. Finally, the proposed method able to classify the sleep disorders of the patients into four classes i.e., healthy, insomnia, SDB, and RBD.

We organize the rest of this paper as follows. Section 2 presents a review of related works. Section 3 describes the materials and the proposed method includes the detail description of the data used in this work, the proposed pre-processing, spectral features extraction, sleep stage detection, the assessment of sleep quality, and classification of sleep disorders patients. In Section 4, we present the results and discussions of the proposed method in classifying sleep disorders. Section 5 is dedicated to the conclusion and further works.

## 2. Related Works

Spectral analysis is widely used in HRV computation for detecting the sleep stage. Typically, HRV is distinguished into three frequency bands, i.e., very low frequency (VLF), low frequency (LF), and high frequency (HF). A study [10] has investigated that the computation of the nonlinear HRV indexes needs even longer segments (at least five minutes), and it may reduce the resolution of the estimated sleep staging results. On the other hand, the Pan-Tompkins algorithm has been widely used for the pre-processing stage of the ECG signal [11] due to its effectiveness in detecting the position of the QRS complex. Accurate detection of the QRS position is a crucial factor in all automatic ECG-based systems, especially in noisy ECG signals. Separating the noise from the ECG signal without destroying the

QRS waveform is a complicated process due to noise is usually broadband and overlaps to the QRS complex. To address this issue, most of the ECG-based systems typically perform filtering. However, it causes the amplitude of QRS's peaks are decreased and degrades the performance [12].

A study [13] reported that there is a significant difference between HRV power and sleep stage condition. The power associated with LF is higher in REM sleep than in NREM sleep, while the power associated with HF is significantly higher in NREM sleep than in REM sleep. However, the standard scoring procedure is necessarily enhanced due to the transition from NREM sleep to REM sleep (and it is vice versa) that may occur suddenly. Another study found that the autonomic changes of HRV may occur before electroencephalographic modifications and show the shift from NREM sleep to REM sleep [14]. In addition, the frequency gaps between REM sleep and NREM sleep are different within the whole sleep episode, but the specific tendencies are observable in the autonomic control of HRV during the night sleep [13].

According to Long, et al. [15], due to HRV apply fixed boundaries for specifying the frequency bands, it may fail to reflect certain aspects of ANS activity accurately. It may limit their discrimination power, e.g., in sleep and wake classification. Therefore, the adapt HRV spectral features are implemented to discriminate the power in classifying sleep and wake. The results showed that the combination between adapted HRV spectral features and other selected HRV non-spectral features significantly improve the overall classification performance, including the sleep and wake. Nevertheless, the overlapped part of the spectrum components will influence the computed features for both bands (LF and HF). It may have an impact on the decreasing accuracy of the classifier. Therefore, a more harmonious method is needed for defining a threshold that is used to separates the two bands.

A study [16] proposed to detect the sleep stage to analyze the sleep condition of the patient. Sleep stage detection is a standard way to analyze sleep. Some works have developed an automatic sleep stage detection using Electroencephalogram (EEG), Electromyogram (EMG), and Electrooculogram (EOG). However, since the complexness of the recording process and signal analysis of those signals, it makes those methods are not recommended to be implemented as a portable system of home appliances. To this end, the use of the ECG signal for detecting the sleep stage is lately massively explored. The diverse feature extraction techniques and machine learning classifiers have been applied in classifying the sleep stage [17–19]. However, most of them just observed some of the sleep stages. In this paper, we observe all sleep stages (i.e., the wakefulness, light sleep, deep sleep, and REM sleep). Structurally, the automatic sleep stage system consists of a preprocessing phase and feature extraction phase. The results of this sleep stage system are used to compute sleep quality features.

Some research [20,21] has used ECG in assessing sleep quality. In the back-end phase, a multi-class Support Vector Machine (SVM) classifier was used to classify sleep quality [22]. This approach presents a good promising in terms of sleep efficiency index, delta-sleep efficiency index, and sleep onset latency. The work of [23] also has investigated a binary classification of sleep stages (sleep-wake stages), and sleep efficiency estimation using the ECG signal. This system achieved an average error of 4.52% for 12 features input and 4.64% for ten features input. Furthermore, the difference between healthy subjects and Obstructive Sleep Apnea (OSA) patients in terms of sleep quality index has been considered by [24] and was developed by [25] on their automatic sleep quality system. It proves that the ECG signals can be used to obtain the sleep quality features.

Each sleep disorder has different characteristics of related sleep quality parameters. Most of the conventional methods used the Pittsburgh Sleep Quality Index (PSQI) to evaluate sleep quality [26]. The PSQI is a self-report questionnaire that assesses sleep quality over a one-month time interval. In PSQI, several questions are related to the psychometric properties of sleep quality. Typically, the International Classification of Sleep disorders (ICSD-3) requires the PSG method for detecting primary clinical sleep disorders. However, some works use the Brief Insomnia Questionnaire (BIQ) to diagnose insomnia [27], and REM Sleep Behavior Disorder Screening Questionnaire (RBDSQ) to diagnose RBD [28]. According to ICSD-3, RBD included in the parasomnias disorders. There are ten questions to assess the various aspects of sleep behavior. The higher score of RBDSQ will be associated with

RBD. However, the reliability and validity of those questionnaires still need further evaluation for some patients. The evaluation result shows the RBDSQ is invalid for Parkinson's disease patients [29].

Even though several works have applied a minimal physiological signal to detect sleep conditions, but only a few works have applied it to diagnose and treat the various sleep disorders instead of the use of a multichannel signal (PSG). A study [30] has developed an algorithm for detecting the sleep arousal. By using K-nearest Neighbours (KNN) classifier, this method achieved averagely 79% of sensitivity, 95.5% specificity, and 93% accuracy. However, the types of arousal and sleep disorder are not distinguished.

At the end of this section, we summarize the state-of-the-art of our proposed method as follows:

1. As mentioned above, different from most current ECG-based automatic sleep stage systems that applied a 5-minute epoch to observe the main frequency band of ECG signal, we perform a new pre-processing technique that suitable for 30-seconds epoch without detecting QRS. We take advantage that the proposed method more efficiently to be implemented in an embedded hardware device as a consideration of the complexity requirements and computational cost.
2. A set of efficient ECG signal features (normalized LF and HF) is extracted by analyzing the HRV frequency band of Power Spectrum Density (PSD) using a Hanning window with the welch method, which is then used to identify the sleep stages.
3. All sleep stage conditions are observed to patients and non-patients subjects. It is an essential factor for a robust sleep stage system.
4. Since the proposed method present an effective and efficient in classifying sleep disorders of the elderly, we expect this method could be used as a general framework in modeling sleep disorders and become a fundamental model for future research. Moreover, we expect that our proposed method can contribute to the ICSD-3 study and aim as a new alternative for diagnosing the sleep disorders, besides using the questionnaire-based method, such as PSQI, BIQ, and RBDSQ.

## 3. Materials and Proposed Methods

The proposed method consists of five-stages, as shown in Figure 1. The first stage is the pre-processing of the ECG signal. Afterward, we extract the spectral features from the ECG signal in the second stage. The third stage is the sleep stage detection process by applying the DTB-SVM. The assessment of sleep quality performs in the fourth stage. In the final stages, sleep disorder classifies using an ensemble of bagged tree classifier. We use Matlab software for all computations in the proposed method.

### 3.1. Data Description

In this work, we concentrated on healthy subjects and the most common elderly's sleep disorders (i.e., insomnia, SDB, and RBD). We used the Cyclic Alternating Pattern (CAP) [31] sleep data. CAP data contains various physiological sleep conditions, non-sleep disorders (or healthy), and sleep disorders, such as Bruxism, Insomnia, Narcolepsy, Epilepsy, PLMS, RBD, and SDB. The average age of subjects is more than 60 years. It means that CAP data represents the age of the elderly.

In the CAP data, the PSG recordings contain the data of EEG channels, EOG channels, EMG signals, respiration signals, and ECG signals. We only use the ECG signals from each subject to the experiments. Based on R&K rules, each epoch (30-seconds) is label into six sleep stages (i.e., wake, NREM 1-4, and REM) [32]. We then re-label each epoch according to the American Academy of Sleep Medicine (AASM) guidelines [33]. The labels are wakefulness, light sleep (a combination of NREM 1&2), deep sleep (a combination of NREM 3&4), and REM sleep. On each epoch, we evaluate 51-subjects that consist of 23–42 years old sixteen healthy subjects (7 males and 9 females), 47–82 years old nine insomnia patients (4 males and 5 females), 65–78 years old four SDB patients (4 males), and 65–78 years old twenty-two RBD patients (19 males and 3 females). Table 1 shows detail information about the total number of epoch in terms of each stage and subject.

**ECG SIGNAL**

**1ˢᵗ stage: Pre-processing**
- Filtering
- R-peak detection
- Interpolating R-R intervals

**2ⁿᵈ stage: Spectral features extraction**
PSD by hanning window with welch method

**3ʳᵈ stage: Sleep stage detection**
Decision-Tree-Based Support Vector Machine (DTB-SVM)

**4ᵗʰ stage: Assessment of sleep quality**
- Total Time in Bed (TIB)
- Total Sleep Time (TST)
- Sleep Onset Latency (SOL)
- Sleep Efficiency (SE)
- % Wakefulness
- % Light sleep
- % Deep sleep
- % REM sleep

Feature Evaluation and Comparison using one-way ANOVA and post hoc Scheffe test

**5ᵗʰ stage: Classification of sleep disorders**
Ensemble of Bagged Tree

**Figure 1.** General scheme of the proposed method for automatic sleep disorders classification.

**Table 1.** Detailed information about the total number of epoch.

| Subject | Sleep Stage | | | |
| | Wakefulness | Light Sleep | Deep Sleep | REM Sleep |
|---|---|---|---|---|
| **Healthy** | 400 (8.45%) | 2080 (43.95%) | 1185 (25.05%) | 1067 (22.55%) |
| **Insomnia** | 1114 (34.93%) | 1227 (38.48%) | 437 (13.71%) | 411 (12.88%) |
| **SDB** | 196 (20.31%) | 473 (49.02%) | 231 (23.94%) | 65 (6.73%) |
| **RBD** | 1322 (23.55%) | 2084 (37.12%) | 1330 (23.69%) | 878 (16.64%) |

### 3.2. Pre-processing

The pre-processing stage is a key process of this proposed method. In this paper, we perform a pre-processing technique that uses 30-seconds epoch of time-domain ECG signal without detecting the QRS positions. This technique consisted of three phases, as shown in Figure 2.

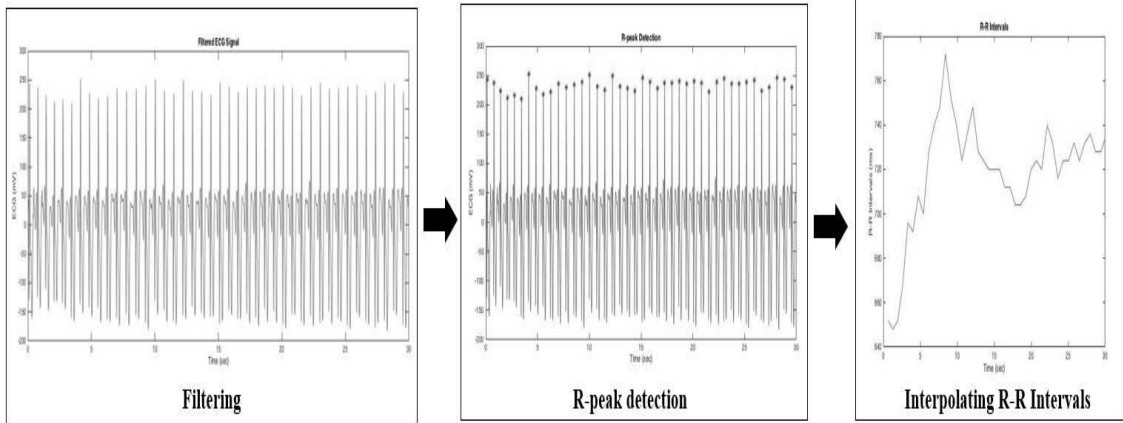

**Figure 2.** Example of pre-processing for first 30-seconds epoch from patient-1 with insomnia.

We present a 30-seconds epoch as a standard recording period for the sleep stage [32,33]. It assumes that the lowest frequency of the 30-seconds epoch is still possible to reach the VLF band power of the HRV spectrum. As shown in Figure 2, we show the first 30-seconds epoch of the ECG signal of patient-1 with insomnia. The process is summarized as follows:

1. Removing the noise using a combination of the band-stop filter and moving average filter.
2. Applying the simple R-peak detector using a 70% threshold from the maximum amplitude of the ECG signal to detect the R-peaks location and use it as a threshold.
3. Interpolating the R-R intervals in the time domain using a cubic spline and re-sampled it at 2.5 Hz. A time-series signal should be re-sampled using frequency sampling at least two-times of the considered maximal frequency. It aims to estimate the HRV spectral (maximum HF band power is 0.4 Hz) for satisfying the Nyquist-Shannon sampling theorem.

### 3.3. Spectral Features Extraction

In the second stage, the spectral features of the ECG signals are extracted. Firstly, we apply the PSD to represent the R-R intervals of the ECG signal to the frequency domain. Specifically, we use the Fast Fourier Transform (FFT)-based welch method to transform from the time domain into the frequency domain. Welch method is a non-parametric technique (includes the periodogram) that provides an excellent resolution and estimation of the spectrum calculation. Then, to reduce spectral leakage on the resulting spectrum, we apply the Hanning window on each epoch. Furthermore, we perform the HRV analysis to the PSD representation. The first 30-seconds epoch of PSD representation (patient-1 with insomnia) is shown in Figure 3. The three primary frequency bands of HRV spectral are computed according to the main frequency band of the HRV spectrum. The HRV spectral primary frequency consists of VLF ranges 0.003–0.04 Hz, LF ranges 0.04–0.15 Hz, and HF ranges 0.15–0.4 Hz. There were few studies explore the significance of VLF fluctuations, and it is still ongoing. While LF fluctuation represents the activity of the parasympathetic and the sympathetic nervous system, and HF fluctuation only reflects the activity of the parasympathetic nervous systems.

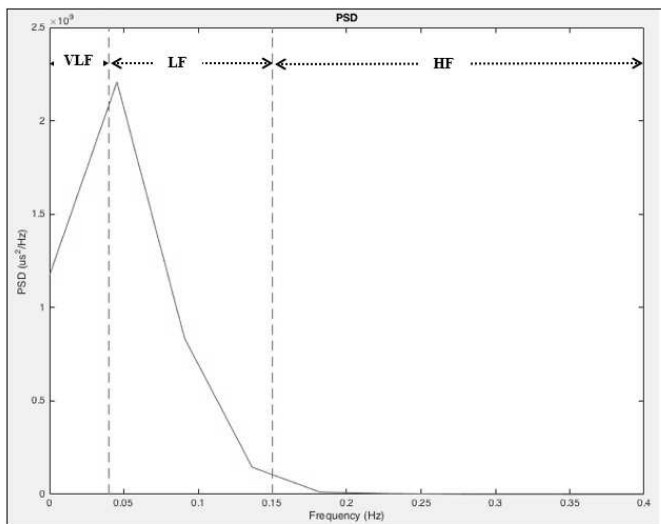

**Figure 3.** Example of Power Spectrum Density (PSD) for first 30-seconds epoch from patient-1 with insomnia.

Typically, the band power measurement of VLF, LF, and HF are represented in absolute values of power ($ms^2$). Since the variation of normalized LF and normalized HF could reflect the balanced and controlled behaviors of the ANS [34], both of them are considered as a good discriminator in distinguishing the sleep stage. Hence, we use the normalized LF and normalized HF feature to recognize the sleep stage. The LF and HF in the normalized unit represent the absolute value of LF and HF in the total distribution of power in spectrum analysis. Thus, the normalized LF and normalized HF band power describe the relative value of LF and HF in proportion to the total spectral power (TSP) minus the VLF band power, as formulated in Equations (1) and (2). The TSP is the total spectral power of HRV (up to 0.4 Hz).

$$Normalized\ LF = \frac{LF}{TSP - VLF} \tag{1}$$

$$Normalized\ HF = \frac{HF}{TSP - VLF}. \tag{2}$$

The variation of obtained HRV from ECG signals is associated with the ANS activity for distinguishing the sleep stage. HRV represents the changes between beat-to-beat variations in the time intervals of the heart rate, called inter-beat or R-R intervals. Furthermore, time-domain describes quantifying the amount of measured HRV in the R-R intervals, such as Standard Derivation of NN Intervals (SDNN) and Root Mean Square of Successive R-R Interval Differences (RMSSD). The burst of parasympathetic and sympathetic can increase the R-R intervals of the heartbeat. Higher values of the SDNN indicate the wakefulness and REM sleep. Thus, the LF band power makes a significant contribution to SDNN. It implies that the parasympathetic and sympathetic HRV measurement is sensitive to SDNN, while the HF band power correlates to RMSSD. Therefore, parasympathetic HRV measurement is sensitive to RMSSD but it not across to the sleep stage [35,36]. A study [37] investigated a certain degree of HRV reduction using SDNN and RMSD. The result shows that there are no significant differences between young and elderly subjects in terms of the linear scale-invariant correlations, nonlinear scale-invariant correlations, the fractal measure of directionality, and nonlinear fractal measure.

### 3.4. Sleep Stage Detection

In this stage, the DTB-SVM is used to detect the sleep stage. DTB-SVM is a combination of SVM and Decision Tree (DT) [38]. The learning function form of NN and SVM is statistically the same. A single hidden layer of NN uses the same model with an SVM. SVM is a machine learning

algorithm that uses the structural risk minimization principle for estimating the objection function. It is performed by minimizing the upper bound of the generalization error. The SVM attempts to seek an optimized linear division, that is, construct a hyperplane that is then used to separate the classes. The general equation of SVM in defining the hyperplane *y(a)* is given in Equation (3), where, *w* is the support vector, *a* is the input vector, and *b* is the bias term.

$$y(a) = w.a + b \tag{3}$$

According to the binary SVM rules that (*x-1*), where *x* is a class problem, for classifying four sleep stages conditions, we use three SVMs. Firstly, we trained the SVM using as input the normalized LF and normalized HF features. It aims to find the optimal hyperplane with the maximum margin (*m*), where the margin can be calculated by dividing the integer two with the absolute function of support-vector. Then, the DT is used to find the maximum margin of each SVM until the four-targeted conditions (i.e., wakefulness, light sleep, deep sleep, and REM sleep) are reached. The detailed scheme for DTB-SVM is illustrated in Figure 4. We evaluate the DTB-SVM performance using the cross-validation method with k = 5.

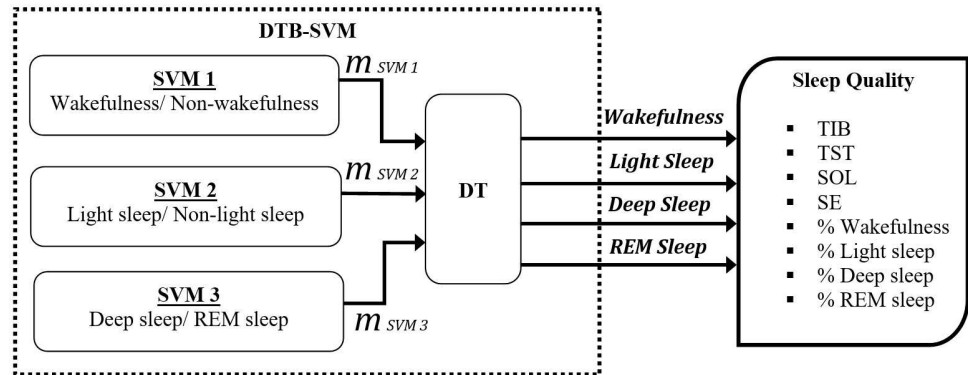

**Figure 4.** General scheme of Decision-Tree-Based Support Vector Machine (DTB-SVM).

*3.5. Assessment of Sleep Quality*

The fourth stage is the assessment of the sleep quality of the subjects. This stage conducts in two steps. The first step is the measurement of sleep quality based on the result of sleep stage detection (previous stage). We use eight sleep quality parameters. In the second step, we analyze the eight parameters using one-way ANOVA and post hoc Scheffe test and select the parameters that have the most significant differences (we called them as the sleep quality features).

Table 2 presents the differences characteristic of the sleep disorders between patient and healthy subjects based on eight sleep quality parameters, i.e., Total Time in Bed (TIB), Total Sleep Time (TST), Sleep Onset Latency (SOL), Sleep Efficiency (SE), and the percentage of each sleep stages (i.e., wakefulness, light sleep, deep sleep, and REM sleep) of insomnia, SDB, and RBD disorders.

**Table 2.** Sleep quality parameters of insomnia, Sleep-Disordered Breathing (SDB) and REM Behavior Disorder (RBD) disorders.

| Features | Insomnia [39] | SDB [40,41] | RBD [41,42] |
|---|---|---|---|
| Total Time in Bed (min) | Decrease | Decrease | Decrease |
| Total Sleep Time (min) | Decrease | Decrease | Decrease |
| Sleep Onset Latency (min) | Increase | Increase | Increase |
| Sleep Efficiency (%) | Decrease | Decrease | Decrease |
| Wakefulness (%) | Increase | Increase | Increase |
| Light sleep (%) | Increase | Increase | Increase |
| Deep sleep (%) | Decrease | Increase | Decrease |
| REM sleep (%) | Decrease | Decrease | Decrease |

The details information of each parameter is described as follows [43–45]:

1. TIB is the total investigation time or the total in-bed duration (in minutes). TIB has a clinical significance for diagnosing sufficient sleep.
2. TST is the total sleep duration or total non-wake conditions (in minutes). TST has a relation for diagnosing the effects of medications, sleep deprivation, and medical condition.
3. SOL is the duration time from the wake condition until getting the first non-wake condition (in minutes). SOL represents sleep time habits.
4. SE is the ratio of total sleep duration (TST) and the total in-bed duration (TIB) (in percentage). In normal sleep conditions, it should at least 85% of TIB. SE represents how well the subject slept.
5. The percentage of wakefulness stage is used to measures awake condition.
6. The percentage of light sleep stage is associated with the transition between being awake and asleep. The increasing percentages of light sleep indicate the patient has a sleep disorder. Typically, the percentage of light sleep is around 55% of the total sleep duration for normal sleep conditions.
7. The percentage of deep sleep stage is associated with the rebound sleep and side effect of medications. SDB disorders are indicated by increasing the percentages of deep sleep [46]. The normal percentage of deep sleep is around 20% of total sleep for normal sleep conditions.
8. The percentage of the REM sleep stage is sensitive to the effect of medications and sleep deprivation. Nevertheless, the REM sleep stage remains approximately 25% of the total sleep in normal sleep conditions. The increasing percentages of the REM sleep indicate a recovery of sleep deprivation.

We engage the one-way ANOVA [47] and post hoc Scheffe test [48] to select the most significant differences in the sleep quality parameters. This step is an essential part of an automatic classification system because the selected features are used to distinguish sleep disorders. One-way ANOVA defines whether one or more sleep disorder parameters are significantly different from each other based on an individual parameter. A null hypothesis of one-way ANOVA describes that the mean (average) of all sleep disorder parameters is equal to the mean (average) of an individual parameter of sleep quality parameters. Otherwise, an against the hypothesis of one-way ANOVA described that at least one the mean (average) of the sleep disorder parameter is different. Furthermore, the mean (average) of the sleep disorder parameters in different sleep disorders data are determined by the post hoc Scheffe test.

### 3.6. Classification of Sleep Disorders

In this fifth stage, we use an ensemble of bagged tree classifier for classifying sleep disorders, namely, healthy, insomnia, SDB, and RBD. An ensemble of bagged tree classifier is a combination of the bagging algorithm and decision tree classifier [49]. The ensemble method is a machine learning technique that combines multiple machine learning classifiers that aims to improve the classification performance, robustness, and reduced over-fitting problem [50]. The Bagging algorithm (or bootstrap aggregation) is one of the most common ensemble methods [51]. The Bagging algorithm is the most

accurate and efficient ensemble method compared to boosting and random forest because it can reduce a high variance of the algorithm, such as the decision tree algorithm [52]. Thus, the Bagged Tree algorithm is a bootstrap procedure implemented in the decision tree (DT) algorithm. The general scheme of the ensemble of bagged tree classifier illustrates in Figure 5. As shown in Figure 5, sleep quality data that consists of healthy, insomnia, SDB, and RBD disorders are used as the training data. Then, we divide this training data randomly into *n* new training data (random subset). Each random subset is used to train one DT classifier. Each DT classifier consists of a root, interior nodes, and leaf nodes. The interior nodes correspond to the attributes, and the leaf nodes correspond to decision results.

For classification problems, the predicted class for an observation is the class that yields the largest weighted average of the class posterior probabilities (classification score) computed using selected trees only. Let *H(x)* be an instance for the DT's output probability distributions $h_i(x, c_j)$, where $i = 1...n$ is the number of DT classifier, and $c_j, j = 1...k$ is the class labels, which is the estimated posterior probability of class $c_j$ given observation *x* using tree *i*. Then, the result aggregation *H(x)*, for instance *x* is obtained using the majority vote. The predicted class is the class that yields the largest weighted average, as expressed in Equation (4). Ensemble of bagged tree computes the optimal result aggregation *H(x)* from each DT classifier. The final prediction obtained from the decision of result aggregation. The sleep disorders determined according to the final prediction of the ensemble of bagged tree classifier.

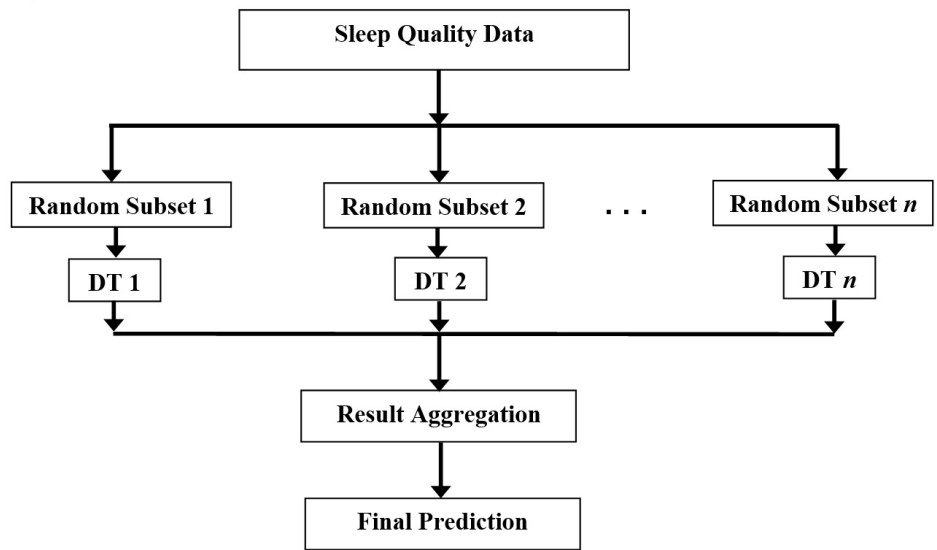

**Figure 5.** General scheme of ensemble of bagged tree classifier.

$$H(x) = arg_{cj}max \sum_{i=1}^{k} h_i(x, c_j) \tag{4}$$

To create the decision trees, we use the standard the Classification and Regression Trees (CART) algorithm. In CART, all input data are examined to all possible binary splits on every predictor. At each node, it will scan over every possible threshold split for every feature, calculate the information gain for each of these different splits, and then choose the split that yielded the highest information gain. The information gain is the number of data points that lead to that node (on the left/right side of the threshold) with the classification, divided by the total number of data points. A split might lead to a child node having too few observations (less than the minimum leaf size parameter). To avoid this, we use a split that yields the best optimization criterion subject to the minimum leaf size constraint,

namely "Gini impurity". Gini evaluates how many observations of each class would be split into each child node, expressed as follows:

$$Gini = 1 - \sum_{j=1}^{k} [p(c_j|t)]^2, \tag{5}$$

where $p(c_j|t)$ denotes the posterior probability of observation belonging to class $c_j$ at a given node $t$.

## 4. Results and Discussion

In this section, we assess the effectiveness of the proposed method for classifying sleep disorders. The performance evaluation and comparison between our proposed method and AASM guidelines included some related clinical study are presented in Section 4.1. In order to describe the implementation planning in an embedded hardware device are provided in Section 4.2.

### 4.1. Experimental Result

In this work, DTB-SVM is used to detect the sleep stage based on spectral features of the ECG signal. Tables 3–6 present the confusion matrix of the sleep stage detection for health and sleep disorders subjects. These matrices represent the performance of the DTB-SVM in terms of true-positive (TP), true-negative (TN), false-positive (FP), and false-negative (FN). TP presents the number of sleep stages that labeled correctly and TN for the number of sleep stages that correctly identified as not correspond to the sleep stages. FP indicates the number of sleep stages that incorrectly labeled. FN denotes the number of sleep stages that unidentified in sleep stage classes. Furthermore, the TP, TN, FP, and FN are used to evaluate the performance of sleep stage detection, such as specificity, sensitivity, and overall accuracy.

The main diagonals of each confusion matrix denote the TP values. A shown in Tables 3–6, the TP values are the highest value on each row and column. It indicates that the proposed method able to detect the sleep stages accurately. In more particular, the specificity, sensitivity, and overall accuracy of the sleep stage detection for 16 healthy subjects achieve 96.36%, 84.63%, and 90.42%, respectively. Table 3 shows that deep sleep is well classified, while wakefulness tends to be miss-classified than the others. It is because wakefulness and light sleep have a similar LF power level for healthy subjects so that they are difficult to distinguish. The lowest value of the LF power level occurs during the deep sleep, while the HF power level during wakefulness is comparable with deep sleep and REM sleep, but different from the light sleep [53].

The specificity, sensitivity, and overall accuracy of the sleep stage detection for nine patients with insomnia achieve 99.15%, 97.16%, and 97.55%, respectively. Table 4 shows that the best classification is provided by light sleep, matching with the PSG scoring. Then, the specificity, sensitivity, and overall accuracy of the sleep stage detection for four patients with SDB are 96.79%, 77.85%, and 89.89%, respectively. The specificity, sensitivity, and overall accuracy of the sleep stage detection for 22 patients with RBD achieved 99.54%, 97.92%, and 98.57%, respectively. Tables 5 and 6 show that deep sleep and REM sleep tend to be miss-classified than the others, while wakefulness and light sleep are well classified for patients with SDB and RBD. It is due to some of the REM condition cannot be detected in patients with SDB and RBD. SDB and RBD distinguish wakefulness and light sleep better than healthy subjects. It is because hypopnea or apnea may have affected the pulse during sleep. RBD patients frequently have SDB. The hypopnea or apnea index (AHI) of SDB and RBD patients is higher than 15. Moreover, RBD is an intriguing parasomnia characterized by repeated episodes of dream enactment behavior and REM Sleep Without Atonia (RSWA). RSWA is characterized by increased tonic muscle activity. Therefore, it could be said that RSWA also influences a diagnostic feature of RBD in REM sleep [54]. For classifying the sleep stage of all subjects, the DTB-SVM achieved an average classification specificity, sensitivity, and overall accuracy of 98.31%, 91.84%, and 95.06%, respectively. Obviously, these results are a great foundation to be used in sleep disorders classification.

**Table 3.** Confusion matrix for healthy subjects.

| | | Automatic Scoring | | | |
| | | Wakefulness | Light Sleep | Deep Sleep | REM Sleep |
|---|---|---|---|---|---|
| | wakefulness | **238** | 155 | 0 | 0 |
| PSG | light sleep | 120 | **1873** | 0 | 0 |
| Scoring | deep sleep | 0 | 0 | **1185** | 0 |
| | REM sleep | 0 | 0 | 169 | **894** |

**Table 4.** Confusion matrix for patients with insomnia.

| | | Automatic Scoring | | | |
| | | Wakefulness | Light Sleep | Deep Sleep | REM Sleep |
|---|---|---|---|---|---|
| | wakefulness | **893** | 6 | 35 | 0 |
| PSG | light sleep | 0 | **1145** | 0 | 0 |
| Scoring | deep sleep | 19 | 3 | **432** | 0 |
| | REM sleep | 0 | 7 | 0 | **321** |

**Table 5.** Confusion matrix for patients with SDB.

| | | Automatic Scoring | | | |
| | | Wakefulness | Light Sleep | Deep Sleep | REM Sleep |
|---|---|---|---|---|---|
| | wakefulness | **1372** | 0 | 0 | 0 |
| PSG | light sleep | 0 | **2329** | 0 | 0 |
| Scoring | deep sleep | 0 | 0 | **1418** | 20 |
| | REM sleep | 51 | 1 | 15 | **898** |

**Table 6.** Confusion matrix for patients with RBD.

| | | Automatic Scoring | | | |
| | | wakefulness | Light Sleep | Deep Sleep | REM Sleep |
|---|---|---|---|---|---|
| | wakefulness | **332** | 0 | 0 | 0 |
| PSG | light sleep | 0 | **620** | 0 | 0 |
| Scoring | deep sleep | 0 | 0 | **324** | 41 |
| | REM sleep | 0 | 0 | 106 | **31** |

Table 7 presents all sleep quality parameters for each subject. By paying attention to the standard deviation, most of the value of sleep quality parameters (for the patients with insomnia, SDB, and RBD compared to healthy subjects) accordance to the characteristics of clinical studies, as shown in Table 2. Furthermore, one-way ANOVA is used to obtain the sleep quality parameters. Then, we use a post hoc Scheffe test to compute the significant differences between two sleep disorder parameters. As shown in Table 7, all parameters of sleep quality (except the TST, percentages of light sleep, and percentages of deep sleep) present statistically significant differences in all sleep disorders at the level of significance ($p$) of 0.05. It indicates that TIB, SOL, SE, percentages of wakefulness, and percentages of REM sleep are the crucial parameters for identifying the elderly's sleep disorders.

Figure 6 shows the trained model of the ensemble of bagged tree classifier in classifying four classes of sleep disorders: healthy subjects, patients with insomnia, SDB, and RBD. The SOL is used to determine patients with insomnia. We found that most of the patients have a higher duration of SOL (above 36.5 min) than others feature. It corresponds to a recent clinical study result [55] that reported elderly patients with insomnia (65 years or older) have more than 30 min of quantitative SOL criteria. Moreover, AASM also has treatment goals to reduce the SOL of insomnia patients with at least lower than 30 min [56]. This treatment aims to improve sleep quality.

Then, we determine that a healthy subject is a subject that has the duration of SOL below 36.5 min and the percentage of wakefulness below 17.505%. It corresponds to a recent clinical study that

estimates the average SOL of healthy people (aged 60 years or older) is around 19 min [57], and we obtained below 36.5 min.

**Table 7.** One-way ANOVA and post hoc Scheffe test to evaluate the importance of sleep quality parameter.s

| | Healthy | Insomnia | SDB | RBD | *p* Value |
|---|---|---|---|---|---|
| **TIB** | 147.63 ± 101.17 [4] | 159.17 ± 105.58 [4] | 181.75 ± 127.33 [4] | 27.98 ± 22.53 [1,2,3] | 0.000016 |
| **TST** | 136.44 ± 102.57 | 108.5 ± 96.25 | 140.25 ± 86.33 | 110.93 ± 66.43 | 0.75 |
| **SOL** | 11.56 ± 7.23 [2] | 85.55 ± 81.07 [1,3,4] | 10 ± 7.52 [2] | 14.10 ± 7.31 [2] | 0.000011 |
| **SE** | 84.64 ± 5.23 [2] | 64.15 ± 17.28 [1] | 71.37 ± 11.61 | 74.69 ± 11.8 | 0.00083 |
| **% Wakefulness** | 5.64 ± 5.23 [2,3,4] | 33.1 ± 17.46 [1,3,4] | 20.54 ± 8.34 [1,2,4] | 23.87 ± 11.8 [1,2,3] | 0.0000017 |
| **% Light sleep** | 40.64 ± 7.36 | 36.85 ± 11.03 | 48.13 ± 18.55 | 37.32 ± 10.27 | 0.23 |
| **% Deep sleep** | 24.29 ± 6.19 | 16.20 ± 5.22 | 20.56 ± 10.11 | 22.99 ± 11.73 | 0.19 |
| **% REM sleep** | 19.72 ± 4.61 [2,3,4] | 11.10 ± 4.98 [1,3] | 2.67 ± 2.27 [1,2,4] | 14.38 ± 5.33 [1,3] | 0.00000046 |

*Note.* Values are expressed as mean ± standard deviation. [1] significantly different from healthy subjects; [2] significantly different from patients with insomnia; [3] significantly different from patients with SDB; [4] significantly different from patients with RBD.

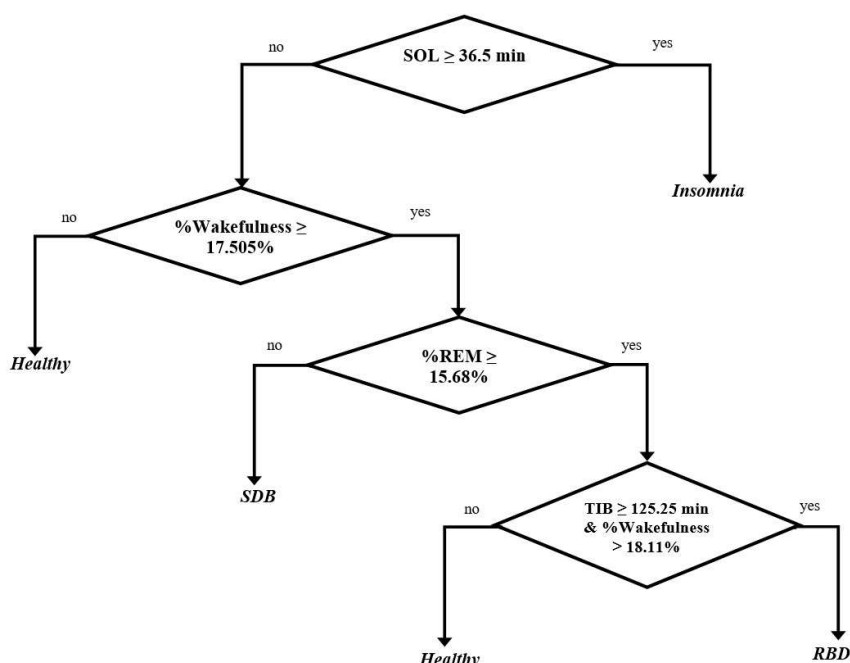

**Figure 6.** The trained model of the ensemble of bagged tree classifier in classifying four classes of sleep disorders.

Furthermore, we found the specification of the duration of SOL, the percentage of wakefulness, and the percentage of REM sleep to determine the SDB patients. The specification describes as follows.

1.　The longest duration of SOL for SDB patients was below 36.5 min It corresponds to a clinical study that estimated the average longest duration of SDB patients was around 9.5 min [58], and we obtained below 36.5 min.
2.　The percentage of wakefulness was above 17.505%.
3.　The longest percentage of REM was below 15.68%. It corresponds to a clinical study that estimated the prevalence percentage of REM in the SDB patients (such as OSA patients) was around 13.5% [59], and we obtained below 15.68%.

Other parameters such as the duration of SOL, the percentage of wakefulness, the percentage of REM sleep, and the duration of TIB were used to determine RBD patients with the specifications as follows.

1.  The longest duration of SOL for RBD patients was below 36.5 min. It corresponds to a related clinical study that estimated the average duration of SOL in 8 RBD patients was around 11.1 min [60], and we obtained below 36.5 min.
2.  The percentage of wakefulness was above 17.505%.
3.  The longest percentage of REM was above 15.68%. It corresponds to a related clinical study that estimated the characteristics percentage of REM in 94 RBD patients was around 22.4% [42], and we obtained above 15.68%.
4.  The duration of TIB and the percentage of wakefulness were generated simultaneously. Thus, the RBD patient is the subject that has the duration of TIB above 125.25 min and the percentage of wakefulness above 18.11%. It corresponds to a clinical study that estimated the average duration of TIB in 4 RBD patients is around 452.75 min [41], and we obtained above 125.25 min. In addition, [57] evaluated that healthy subjects have an average percentage of wakefulness around 10.55%, and we obtained below 17.505% and 18.11%.

Furthermore, the confusion matrix and the performance evaluation of the proposed method are presented in Tables 8 and 9, respectively. The highest sensitivity is achieved by RBD (90.0%) with none miss-classification in insomnia and SDB. Conversely, specificity and accuracy of RBD are lowest (82.76% and 86.27%, respectively) than the other. It is because the classification of RBD depends on four sleep quality features, i.e., SOL, wakefulness, REM, and TIB. Moreover, the miss-classification of RBD into healthy due to there are two subjects that have a percentage of wakefulness lower than 18.11%, but the TIB is above 125.25 min.

Specificity and accuracy of insomnia and SDB are the highest, but there is one subject on each classified as RBD. Insomnia is expected to be classified by SOL, but there is one subject that has a duration of SOL below 36.5 min. For the same reason, one subject of SDB has a percentage of REM higher than 15.68%. Furthermore, there are healthy subjects classified as RBD. It due to three subjects has a percentage of wakefulness higher than 18.11%, and the TIB is above 125.25 min.

The evaluation performance of the whole proposed method achieved 84.01% of sensitivity, 94.17% of specificity, 86.27% of overall accuracy, and 0.70 of Cohen's kappa. The specificity of 94.17% implies that the proposed method able to accurately distinguish the sleep disorder subjects and non-patients (healthy subjects). The sensitivity of 84.01% shows that the ability to recognize healthy subjects and elderly sleep disorder patients is quite robust. The proposed method also presents a good overall accuracy (86.27%). The Cohen's kappa that represents the inter-rater agreement with PSG scoring achieves 0.70. According to the rule of thumb values of Cohen's kappa [61], 0.70 implied a substantial level of agreement between proposed automatic scoring and actual PSG scoring. The proposed method is reliable as an automatic sleep disorders model and able to become a good foundation for classifying the elderly sleep disorders based on sleep quality features from the 30-seconds epoch ECG signal.

**Table 8.** Confusion matrix of an automatic classification system.

|  |  | Automatic Scoring | | | |
|  |  | Healthy | Insomnia | SDB | RBD |
|---|---|---|---|---|---|
|  | Healthy | **13** | 0 | 0 | 3 |
| **PSG** | Insomnia | 0 | **8** | 0 | 1 |
| **Scoring** | SDB | 0 | 0 | **3** | 1 |
|  | RBD | 2 | 0 | 0 | **20** |

Table 10 shows a comparison of our work with several related studies that also distinguish sleep disorders. The researchers [62] developed an automated detection system of sleep disorders using related events from the EEG signal. This system has two types of events (i.e., arousal events and left & right leg movement events) that detected for discriminating SDB (such as OSA) and sleep-related movement disorders (such as RLS). They used a DT classifier to estimate the total accuracy of the arousal and left and right leg movement events. Based on their experimental result,

this system presented the overall accuracy by 85.02%. The other approach [63] developed an automatic classification system for sleep apnea episodes. This approach used three types of features vectors, i.e., PSD estimation, Principal Component Analysis (PCA), and Partial Least Squares (PLS) scores from the EEG signal. Then, the SVM classifier used for discriminating SDB. This system achieved an overall accuracy of 85%. As mentioned above, both works [62,63] presented good accuracy results. However, both of them have not reported their system performance in more detail, such as the specificity, sensitivity, and Cohen's kappa yet.

**Table 9.** Evaluation performance of each sleep disorder subjects and non-patients (healthy subjects).

|  | Sensitivity (%) | Specificity (%) | Accuracy (%) | Cohen's Kappa (%) |
|---|---|---|---|---|
| Healthy | 81.25 | 93.94 | 89.80 | 0.71 |
| Insomnia | 88.89 | 100.00 | 97.78 | 0.71 |
| SDB | 75.00 | 100.00 | 97.78 | 0.60 |
| RBD | 90.91 | 82.76 | 86.27 | 0.73 |

**Table 10.** Comparison work.

|  | Input Signal | Features Extraction | Classifier | Sleep Disorders | Overall Accuracy |
|---|---|---|---|---|---|
| [62] | EEG | Arousal events and Left & right leg movement events | DT | OSA RLS | 85.02% |
| [63] | EEG | PSD estimation, PCA, and PLS | SVM | SDB | 85% |
| **This work** | ECG | Sleep quality | Ensemble of Bagged Tree | Healthy Insomnia SDB RBD | 86.27% |

As shown in Tables 8 and 9, the proposed method presents an improvement in classifying the healthy subjects and sleep disorders patients in the elderly (i.e., insomnia, SDB, and RBD) using ECG based-sleep quality features as the input and an ensemble of the Bagged tree as the classifier. Moreover, the total accuracy of the proposed method is higher compared to the works of [62,63]. We also take advantage of the ECG signal usage that easy to record even though an untrained user. Thus, the proposed method is easy and efficient to implement in the real hardware system. Even though the number of subjects was limited in this work, but the number of epochs is large enough (over 14,000 epochs). Thus, we argue that it is reliable for modeling an automatic sleep disorder for the elderly based on sleep quality parameters from the ECG signal. We assume that it is advantageous to diagnose and treat the various sleep disorders instead of the use of a multichannel signal (PSG).

Finally, we present the trained model of the ensemble of bagged tree classifier for classifying four classes of sleep disorders: healthy subjects, patients with insomnia, SDB, and RBD, as shown in Figure 6. This model takes a decision tree structure and then applies bagging (bootstrap aggregating) to reduce variance and bias. Due to the decision trees divide the predictors by thresholds, so it does not difference how far is a data point from thresholds. Therefore, most likely outliers will have a negligible effect because the nodes are determined based on the sample proportions in each split region (and not on their absolute values). Thus, we ensure that the trained model of ensemble bagged tree (Figure 6) is universal for the entire elderly population.

*4.2. Implementation Planning*

In the experimental result, the proposed method presents easiness and efficiency in classifying sleep disorders by using the ECG signal alone. It indicates that the proposed method gives a big promise to be implemented in an embedded hardware device. Since sleep disorders may happen randomly during the entire sleeping period, we plan to monitor and process the ECG signal in real-time.

As shown in Figure 1, the proposed method consists of five stages. Real-time monitoring and data processing is performed only for the first three stages (pre-processing, spectral features extraction, and sleep stage detection), while the assessment of sleep quality and the classification of sleep disorders are conducted after the night sleep. Pre-processing and features extraction process is applied to each epoch of the ECG signal. The spectral features (normalized LF and normalized HF) are used as the input of DTB-SVM. Then, by using the transfer function of trained DTB-SVM, the sleep stage label of each epoch is determined. We store the sleep stage label of each epoch in the $d$-dimensional labels, defined as follows:

$$S_i = (l_1, l_2, ..., l_j, l_d), \tag{6}$$

where, $S_i$ is $i^{th}$ patient, $l_j$ is a sleep stage label for $j^{th}$ epoch, and $d$ is the number of total epochs.

## 5. Conclusions

In this paper, an easy and efficient method for the automatic sleep disorders classification in the elderly using ensemble bagged tree has been proposed. In the pre-processing stage, we presented a suitable pre-processing technique for the 30-seconds epoch of the ECG signal. Furthermore, spectral features were extracted using PSD by the Hanning window with the welch method. The sleep stage detection was examined using the DTB-SVM. Then, the sleep quality parameters such as TIB, TST, SOL, SE, and the percentage of each sleep stage were computed and analyzed using ANOVA and post hoc Scheffe test to evaluate and select the most important features of sleep quality parameters. The experiment result has been showed the ensemble of bagged tree classifier based on the sleep quality features is able to discriminate the sleep disorders (insomnia, SDB, and RBD) and healthy effectively by achieving a good specificity, sensitivity, overall accuracy, and Cohen's kappa.

We conclude that it is possible to determine sleep disorders based on sleep quality features from 30-seconds epoch of the ECG signal. This approach is proven reliable in modeling sleep disorders without preoccupied with a multichannel signal of PSG. Moreover, it also easy to be implemented in an embedded hardware device.

On the other hand, atrial fibrillation and other heart rhythm disorders are prevalent in the elderly population. It might have an impact on the HRV analysis. However, HRV able to assess sympathetic and parasympathetic influences on disease states. Hence, in further analysis, HRV can be improved following the intervention, and thus it has the ability to assess autonomic dysfunction in the elderly's heart rhythm disorders, such as atrial fibrillation, bradyarrhythmias, and ventricular arrhythmias. In the future, we interest to observe the autonomic dysfunction in the elderly via HRV intervention.

**Author Contributions:** Theory and conceptualization, E.R.W. and K.T.; data requirement, E.R.W. and K.T.; methodology, E.R.W. and K.T.; software design and development, E.R.W.; validation E.R.W., K.T. and H.T.; formal analysis, E.R.W. and K.T.; investigation, E.R.W.; writing—original draft preparation, E.R.W.; visualization, E.R.W. and K.T.; supervision, K.T. and H.T. All authors have read and agreed to the published version of the manuscript.

**Funding:** This research received no external funding.

**Acknowledgments:** The authors gratefully acknowledge the Physionet for providing the Cyclic Alternating Pattern (CAP) data and the anonymous reviewers for their helpful comments and suggestion.

**Conflicts of Interest:** The authors declare no conflict of interest.

## Abbreviations

The following abbreviations are used in this manuscript:

| | |
|---|---|
| AASM | American Academy of Sleep Medicine |
| AHI | Apnea–Hypopnea Index |
| ANOVA | Analysis of Variance |
| ANS | Autonomic Nervous System |
| BIQ | Brief Insomnia Questionnaire |

| | |
|---|---|
| CAP | Cyclic Alternating Pattern |
| CART | Classification and Regression Trees |
| DT | Decision Tree |
| DTB-SVM | Decision-Tree-Based Support Vector Machine |
| ECG | Electrocardiography |
| EEG | Electroencephalogram |
| ELS | Ensemble Learning Systems |
| EMG | Electromyogram |
| EOG | Electrooculogram |
| FFT | Fast Fourier Transform |
| FN | False-Negative |
| FP | False-Positive |
| HF | High Frequency |
| HRV | Heart Rate Variability |
| ICSD | International Classification of Sleep disorders |
| LF | Low Frequency |
| NN | Neural Network |
| NREM | Non Rapid Eye Movement |
| OSA | Obstructive Sleep Apnea |
| PCA | Principal Component Analysis |
| PLMS | Periodic Leg Movement of Sleep |
| PLS | Partial Least Squares |
| PSD | Power Spectrum Density |
| PSG | Polysomnography |
| PSQI | Pittsburgh Sleep Quality Index |
| R&K | Rechtschaffen & Kales |
| RBD | REM Behavior Disorder |
| RBDSQ | REM Sleep Behavior Disorder Screening Questionnaire |
| REM | Rapid Eye Movement |
| RLS | Restless Leg Syndrome |
| RMSSD | Standard Derivation of NN Intervals |
| RSWA | REM Sleep Without Atonia |
| SDB | sleep-Disordered Breathing |
| SDNN | Standard Derivation of NN Intervals |
| SE | Sleep Efficiency |
| SOL | Sleep Onset Latency |
| SVM | Support Vector Machine |
| TIB | Total Time in Bed |
| TN | True Negative |
| TP | True Positive |
| TSP | Total Spectral Power |
| TST | Total Sleep Time |
| VLF | Very Low Frequency |

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
