# Peer review of "Automatic Sleep Disorders Classification Using Ensemble of Bagged Tree Based on Sleep Quality Features"

_electronics, doi:10.3390/electronics9030512_

Round 1

Reviewer 1 Report

In this work, the authors reported a new ECG signal-based method for sleep disorders classification in the elderly. There are five steps in the proposed classification method: 1) pre-processing, 2) spectral feature extraction, 3) sleep stage detection, 4) sleep quality features assessment, and 5) sleep disorders classification. The proposed method has been validated by the study of 51 subjects with common types of sleep disorders and the specificity of 93.31%, sensitivity of 82.45%, and overall accuracy of 84.31% have been achieved. The test results suggested that a sleep disorder detection and classification could be achieved by analyzing the sleep quality features from 30 seconds epoch of the ECG signal, which make this method a simple and efficient way for sleep disorders identification, compared to conventional methods. The research is well designed and performed, and the paper is clearly presented. The impact of the proposed research is considered to be high. This manuscript can be published in Electronics with minor revision.

  • Sleep disorders may happen randomly during the entire sleeping period, if this method is implemented into the hardware, how does the data processing look like? Is a real-time monitoring and data processing required?
  • It's better to show the sleep disorders classification process by indicating how the ECG signal is processed step-by-step, using the data from one subject as an example.
  • More details are needed for the Spectral Features Extraction process.
  • The performance of the proposed sleep disorder classification should be compared with the other methods reported in literatures to better justify its advantage and value, better to be in a summary table.
  • How are the threshold determined in the decision tree in Figure 4? And How reliable these threshold numbers are? Are they universal for the entire elderly population?
  • For those date that are mis-classified, any analysis on the reasons?
  • What kinds of respiratory event parameters could be used to improve the accuracy of the proposed classification methods? How to obtain these respiratory event parameters?

Author Response

Dear Reviewer,

Firstly, we wish to thank the reviewer who has provided suggestions and comments on our manuscript. In the below, we attached the responses document to the reviewer. We have revised this manuscript very carefully. We fundamentally agree with all the comments made by the reviewers, and we have incorporated corresponding revisions into the manuscript. The point-to-point replies and explanations for all the revisions are listed for easy reference. In the revised manuscript, we marked corresponding revision in colored text (blue). We hope that the revised manuscript can be published in the Special Issue “Applications of Bioinspired Neural Network” of Electronics MDPI Journal following these significant changes.

Thank you,

Best Regards,

Authors

Reviewer 2 Report

This paper showed very interesting methods for automatic sleep staging and diagnosing using DTB-SVM and Ensemble of Bagged Tree classifiers. In addition, 30-sec epoch was used which was valid for sleep staging. With some minor changes, this work can contribute to the use of automatic sleep diagnosis.

Table 3 and related argument: Automatic scoring distinguished deep sleep and REM sleep particularly well. This is because the two sleep stages have significantly different heart rate and autonomic nervous system characteristics than the other stages. (In contrast, wakefulness and light sleep are indistinguishable.) More discussion on this point is needed based on the literature.

Table 5 and related argument: The reason why SDB distinguished wakefulness and light sleep better than healthy people is that hypopnea or apnea may have affected the pulse during sleep. Can you provide AHI information for 4 SDB patients and 22 patients with RBD? (since RBD patients frequently have SDB)

Table 6 and related argument: RBD was likely to have the wrong REM sleep due to the large number of REM without atonia (RWAs) in the REM sleep. More discussion on this point is needed based on the literature.

Table 8 and related argument: Although the number of patients for each disease is small, it is better to show specific data (sensitivity, specificity, accuracy, Cohen) for each disease, not just the whole.

Author Response

(The authors gave the same response as above.)

Reviewer 3 Report

This study proposes a method for automatic classification of sleep disorders, including insomnia, sleep-disordered breathing and REM behavior disorder, based on spectral features of electrocardiographic (ECG) signals. This method consists of five stages, including the pre-processing of ECG signals, the spectral features extraction from ECG signals, the sleep stage detection, the assessment of sleep quality parameters and, finally, the classification of sleep disorders. The authors compare the performance of their method with standard polysomnographic recordings in 16 healthy subjects and 35 patients with sleep disorders selected from the Cyclic Alternating Pattern sleep database (Terzano et al. 2001). The proposed method presents good specificity, sensitivity, accuracy and Cohen’s kappa in the identification of sleep disorders compared to classic polysomnography. The authors conclude that it is possible to identify a number of sleep disorders by assessing sleep quality features detected from ECG signals.

The topic of the study is of interest, but the novelty is quite low. Moreover, the current form of the manuscript is confusing and repetitive, requiring an extensive editing of English language and style. I have a number of further comments to improve the current form of the manuscript:

  • The introduction section is too verbose and not linear. The authors should summarize the text, paying attention to the correct flow of concepts;
  • Several previous studies (e.g. Bonnet et al. 1997; Versace et al. 2003; Long et al. 2014; Shahrbabaki et al. 2016) investigating the same research topic are not included in the text. The authors should improve the background of their study, highlighting limitations of previous findings and the novelty of their research protocol;
  • The paragraph 1.2 in the introductory section presents several methodological information (e.g. Lines 133-146, Page 4) that should be reported in the methods section (section 2. “Materials and Proposed Methods”);
  • The methods section presents background information (e.g. Lines 187-199, Page 6) that should be reported in the introductory section;
  • Table 2 may seem in contrast with results because apparently reports similar sleep quality parameters (all except deep sleep %) in the evaluated sleep disorders. Please clarify;
  • Available data do not support greater efficacy of the proposed method in the identification of sleep disorders compared to clinical scales. Accordingly, the authors should tone down their conclusions (Lines 457-458, Page 14);
  • Given the amount of data, the authors should report results and discussion in two separate paragraphs to facilitate reading and understanding;
  • Atrial fibrillation and other heart rhythm disorders are very common in elderly population. The authors should discuss the possibility to use the proposed method in these people;
  • Respiratory sinus arrhythmia is a physiological phenomenon commonly observed in young people, while is less evident in elderly, thus not supporting the authors’ prospect in the conclusion paragraph (Lines 470-473; Page 15).

Author Response

(The authors gave the same response as above.)

Round 2

Reviewer 2 Report

The authors have responded appropriately to the comments.

Author Response

Dear Reviewer,

We pleasured have received a positive evaluation. We also expressed our appreciation for your helpful comments and suggestions.

Thank you so much,

Best Regards,

Authors

Reviewer 3 Report

The authors have improved the quality and overall scientific impact of the study. I would only suggest to reduce the length of the introduction and ask for revision of a native english speaker.

Author Response

Dear Reviewer,

We are grateful to the reviewer for your detailed evaluation and suggestion on our manuscript. In this revised manuscript, we have reduced the length of the introduction (revised in section Introduction Page 1 -2).

Moreover, we do our best to improve all writing styles and English language very carefully through discuss with some English speakers and use an automated grammar checker tool. We marked the majority revised of writing styles and English language according to the reviewer’s comment in colored text (blue).  Since the limitation time of the revision’s process, we apologize that we cannot ask to native English speaker for improving our manuscript. However, if this revised manuscript still be deemed insufficient, we are willing to use Specialist English Editing Services that provided from MDPI.

We hope that the revised manuscript can be published in the Special Issue “Applications of Bioinspired Neural Network” of Electronics MDPI Journal following these significant changes.

Thank you so much,

Best Regards,

Authors